# Evaluation of GPM-era Global Satellite Precipitation Products over Multiple Complex Terrain Regions

Yagmur Derin [1], Emmanouil Anagnostou [1,*], Alexis Berne [2], Marco Borga [3],
Brice Boudevillain [4], Wouter Buytaert [5], Che-Hao Chang [6], Haonan Chen [7], Guy Delrieu [4],
Yung Chia Hsu [8], Waldo Lavado-Casimiro [9], Bastian Manz [5], Semu Moges [10],
Efthymios I. Nikolopoulos [11], Dejene Sahlu [12], Franco Salerno [13],
Juan-Pablo Rodríguez-Sánchez [14], Humberto J. Vergara [15] and Koray K. Yilmaz [16]

1   Department of Civil and Environmental Engineering, University of Connecticut, Storrs, CT 06269, USA;
    yagmur.derin@uconn.edu
2   Environmental Remote Sensing Laboratory, École Polytechnique Fédérale de Lausanne, 1015 Lausanne,
    Switzerland; alexis.berne@epfl.ch
3   Department of Land, Environment, Agriculture and Forestry, University of Padova, 35139 Padova, Italy;
    marco.borga@unipd.it
4   Institute of Environmental Geosciences, University of Grenoble Alpes Community, F-38000 Grenoble, France;
    brice.boudevillain@ujf-grenoble.fr (B.B.); guy.delrieu@ujf-grenoble.fr (G.D.)
5   Department of Civil and Environmental Engineering, Imperial College London, London SW7 2AZ, UK;
    w.buytaert@imperial.ac.uk (W.B.); bastian.manz10@imperial.ac.uk (B.M.)
6   Department of Civil Engineering, National Taipei University of Technology, Taipei 10608, Taiwan;
    chchang@ntut.edu.tw
7   Physical Sciences Division of NOAA—Earth System Research Laboratory, Boulder, CO 80301, USA;
    haonan.chen@noaa.gov
8   Disaster Prevention and Water Environment Research Center, National Chiao Tung University,
    Hsinchu 30010, Taiwan; ychsu1978@ntu.edu.tw
9   Servicio Nacional de Meteorología e Hidrología, Lima 10032, Peru; wlavado@senamhi.gob.pe
10  School of Civil and Environmental Engineering, Addis Ababa Institute of Technology, Addis Ababa 1000,
    Ethiopia; Semu.moges@uconn.edu
11  Department of Mechanical and Civil Engineering Florida Institute of Technology Melbourne, Melbourne,
    FL 329016975, USA; enikolopoulos@fit.edu
12  Institute of Disaster Risk Management & Food Security Studies, Bahir Dar University, Bahir Dar 6000,
    Ethiopia; dejenesahlu@gmail.com
13  National Research Council, Water Research Institute, 38930 Brugherio (IRSA-CNR), Italy; salerno@irsa.cnr.it
14  Department of Civil and Environmental Engineering, Universidad de los Andes, Bogotá 111711, Colombia;
    pabl-rod@uniandes.edu.co
15  Cooperative Institute for Mesoscale Meteorological Studies, The University of Oklahoma, Norman,
    OK 73019, USA; humber@ou.edu
16  Department of Geological Engineering, Middle East Technical University, 06531 Ankara, Turkey;
    yilmazk@metu.edu.tr
*   Correspondence: manos@uconn.edu; Tel.: +1-860-486-6806

**Abstract:** The great success of the Tropical Rainfall Measuring Mission (TRMM) and its successor Global Precipitation Measurement (GPM) has accelerated the development of global high-resolution satellite-based precipitation products (SPP). However, the quantitative accuracy of SPPs has to be evaluated before using these datasets in water resource applications. This study evaluates the following GPM-era and TRMM-era SPPs based on two years (2014–2015) of reference daily precipitation data from rain gauge networks in ten mountainous regions: Integrated Multi-SatellitE Retrievals for GPM (IMERG, version 05B and version 06B), National Oceanic and Atmospheric Administration (NOAA)/Climate Prediction Center Morphing Method (CMORPH), Global Satellite Mapping of Precipitation (GSMaP), and Multi-Source Weighted-Ensemble Precipitation (MSWEP),

which represents a global precipitation data-blending product. The evaluation is performed at daily and annual temporal scales, and at 0.1 deg grid resolution. It is shown that GSMaPV07 surpass the performance of IMERGV06B Final for almost all regions in terms of systematic and random error metrics. The new orographic rainfall classification in the GSMaPV07 algorithm is able to improve the detection of orographic rainfall, the rainfall amounts, and error metrics. Moreover, IMERGV05B showed significantly better performance, capturing the lighter and heavier precipitation values compared to IMERGV06B for almost all regions due to changes conducted to the morphing, where motion vectors are derived using total column water vapor for IMERGV06B.

**Keywords:** satellite-based precipitation product; complex terrain; validation

---

## 1. Introduction

To understand and manage water systems under a changing climate and meet the increasing demand for water, a quantitative understanding of the water cycle at a regional to global scale is important. Gaining this understanding requires that precipitation—a fundamental component of the water cycle—be studied in detail. Precipitation is the main trigger of natural hazards like floods, landslides, and avalanches over complex terrain, but the measurement of its frequency and quantity over these regions has been very difficult. The small space-time scales and remarkable rain rates associated with complex terrain stretch the need for improved quantitative precipitation estimates (QPEs) at high spatiotemporal resolution. In short, an understanding of the specific precipitation formation processes and complex patterns resulting from complex topography and associated vertical gradients is greatly needed.

Rainfall measurement can be conducted at the ground surface—for example, by rain gauge and radar networks—or from space with satellite sensors. While rain gauges provide direct physical measurement of surface rainfall, they are susceptible to sources of uncertainty, such as the size of the collector, evaporative loss, wind, siting, and so on [1,2]. Furthermore, the sparseness of gauge networks over complex terrain limits the spatial representation of rainfall variability. Weather radar networks, on the other hand, provide rainfall measurement at high spatial and temporal resolutions. They are also susceptible, however, to certain limitations and errors, especially over mountainous regions, where accuracy tends to degrade as a result of beam overshooting, range effects, beam blockage, and bright band effects [3,4]. In both cases, establishing and maintaining ground-based gauge and radar networks is often cost prohibitive, especially in remote parts of the world or regions with limited financial resources. Thus, the observations that are generally available are over lowlands, leaving an observational gap in global precipitation data over complex terrain, where rainfall is typically characterized by high spatial variability leading to small- to medium-scale hydrological events (such as flash floods). The scarcity of ground-based observations in these regions affects the reliability of hydrological modeling and water resources assessment [5,6].

The difficulties in the representation of spatial rainfall variability from ground-based observations highlights the need to use integrated satellite-based precipitation products (SPPs)—that is, combined infrared (IR) radiances and passive microwave (PMW) precipitation retrievals—because of their ability to represent the space-time variability of rainfall with quasi-global coverage. The great success of the Tropical Rainfall Measuring Mission (TRMM) has accelerated the development of rain retrieval algorithms, such as the Goddard profiling algorithm (GPROF) [7]. The PMW algorithms retrieve precipitation based on the scattering effect in high-frequency channels (equal to or higher than 85 GHz) over land. These estimates are normally characterized by uncertainties, with more pronounced biases during extreme events due to the mean observed ratio between ice aloft and the surface rainfall. This underestimation from satellite sensors is more distinct over mountainous regions [8–10] and is often attributed to the occurrence of heavy precipitation associated with warm-rain processes [11,12].

In short, the indirect nature of the measurement, the spatial heterogeneity of the precipitation fields, the sensor resolution, the sensor sensitivity, and the retrieval algorithms instigates SPPs to be susceptible to significant uncertainty. To take advantage of the strengths of both IR and PMW, the most recent algorithms combine IR and PMW observations to create global-scale, multi-satellite precipitation products. Well-known products include the Integrated Multi-satellitE Retrievals for Global Precipitation Measurement (GPM, IMERG) Early-, Late-, and Final-Run (gauge-adjusted) products. The GPM Core Observatory is an advanced successor to the TRMM satellite, with additional sensors and channels, like the Dual-frequency Precipitation Radar (DPR) and the GPM Microwave Imager (GMI), with capabilities to sense light rain and falling snow [13,14]. Other currently distributed products are the National Oceanic and Atmospheric Administration (NOAA) Climate Prediction Center morphing technique (CMORPH) [15] and gauge-adjusted product [16] and the gauge-adjusted Global Satellite Mapping of Precipitation (GSMaP) datasets produced at the Earth Observation Research Center (EORC) of the Japan Exploration Agency (JAXA) [17,18].

These SPPs have been evaluated for different continental regimes in the past two decades [19,20]. Regional studies have been conducted over the continental United States [21–24], South America [25–28], Europe [29–31], Africa [32–34], and Asia [35,36] by considering various spatial scales, temporal scales, precipitation regimes, and so on. One review of multiple SPPs during the TRMM era to assess their accuracy [37] showed that in regions characterized by complex orography, SPP rainfall detection was weak, and mean error exhibited magnitude dependence. SPPs have, overall, demonstrated large uncertainties for heavy precipitation events [24,38–40].

In light of this literature, it is now well known that SPPs over complex terrain suffer from various sources of uncertainty. However, with the aid of error correction procedures it has been shown that these datasets can still offer important information for numerous hydrometeorological applications. The authors of one study [41] modeled SPP errors over mountainous terrain by using a well-known SREM2D model and showed their error model was able to correct SPPs over complex terrain with better performance at the daily scale than at the 3-hourly scale. A subsequent study [19] sought to understand this by evaluating nine different SPPs (TRMM sensors and algorithms) over nine complex terrains throughout the globe. The present study is an extension of the latter. In it, we evaluate GPM-era sensors and algorithms to gain an understanding of the improvements to them and their ongoing deficiencies.

To address this goal, in this article we evaluate the major global SPPs over multiple regions of the Earth characterized by complex terrain. The products include IMERGV05B, IMERGV06B, CMORPH, GSMaPV07, and Multi-Source Weighted-Ensemble Precipitation (MSWEP) V2.2, and we evaluate them over ten mountainous regions: the Italian and Swiss Alps, the French Cévennes, the western Black Sea region of Turkey, the Colombian Andes, the Peruvian Andes, the U.S. Rocky Mountains, the upper Blue Nile, western Taiwan, the San Francisco Bay area in California, and the Himalayas over Nepal. We conducted the evaluations by taking averages of rain gauge measurements over corresponding SPP native grid resolutions during 2014 and 2015. The performance of the SPPs was assessed using statistical measures and visual comparison methods. Section 2 provides the details of the study area and rain gauges, and Section 3 describes each of the SPPs. Evaluation methodology is presented in Section 4 and the results in Section 5. Section 6 discusses the results and summarizes our conclusions.

## 2. Study Regions and Rain Gauge Networks

As mentioned above, the study areas involve ten data-rich mountainous ground validation sites. All the regions differ in rain gauge network density and time periods of data record. To make this study consistent across different domains, we limit the ground validation datasets to those with daily time resolution, in which in situ data records span two years (2014–2015) (Table 1). To represent elevation effects, we divided certain regions into those with lower-elevation rain gauges and higher-elevation rain gauges. It should be noted that since no universal quality control has been applied to the rain

gauge data, their quality most likely varies, which may affect the comparison of results across the different regions.

**Table 1.** General information on the rain gauge datasets representing the ground validation of the study regions.

| Region | Geographic Extent (Latitude/Longitude) | Number of Gauges (% of Pixel with Multiple Rain Gauges) | Elevation Range (m) |
|---|---|---|---|
| Blue Nile | 36.46°–38.5°N 10.00°–12.85°E | 56 (9.8%) | 1615–3012 |
| Eastern Italian Alps | 10.42°–12.48°N 46.16°–47.20°E | 57 (16.6%) | 212–1950 |
| Swiss Alps | 6.00°–10.50°N 45.80°–47.78°E | 251 (16.6%) | 203–3302 |
| Turkey | 31.48°–34.15°N 40.25°–41.70°E | 25 (0%) | 1–1305 |
| Peruvian Andes | 68.65°–79.93°S 4.96°–17.60°W | 466 (4.2%) | 6–5088 |
| Colombian Andes | 72.38°–78.00°S 0.78°–7.36°E | 97 (4.3%) | 490–3310 |
| Taiwan | 120.50°–120.90°N 23.20°–23.53°E | 34 (85.7%) | 531–2540 |
| Nepal/Himalayas | 85.15°–88.00°N 26.15°–28.75°E | 11 (22.2%) | 497–5600 |
| US Rocky Mountains | 103.10°–123.15°S 31.35°–48.80°E | 1167 (97.4%) | 74–3124 |
| French Cevennes | 3.00°–4.60°N 43.65°–45.40°E | 432 (42.5%) | 1–1567 |
| California | 120.18°–124.11°S 35.38°–40.97°E | 60 (29.2%) | 15–2014 |

Derin and colleagues [19] have described the climatology of each of the ten study regions. They can be briefly characterized as follows:

*Taiwan:* A tropical climate with annual precipitation exceeding 3300 mm, unevenly distributed throughout the year.

*Swiss Alps:* Under the effect of southern circulation from the Mediterranean, which generates intense precipitation on the southern side of the Alps in the autumn.

*Blue Nile:* Widely varying topography that strongly influences climate, from cool highlands to hot deserts, with most precipitation occurring in the wet season (June–September), driven by the intertropical convergence zone (ITCZ) that drives the summer monsoon.

*Eastern Italian Alps:* Steep topographical gradients where the precipitation is primarily attributed to early fall and frontal systems during fall and early winter.

*French Cévennes:* A largely Mediterranean climate, with annual precipitation totals ranging from about 500 mm near the sea to 2000 mm over the mountains.

*Turkey:* In general, affected in the winter by polar air masses of continental origin of cold Siberian high and maritime origin of Iceland low and in the summer by subtropical air masses.

*Himalayas:* Located in the subtropical zone and influenced by the monsoon system, with local circulation dominated by a system of mountain and valley breezes.

*U.S. Rocky Mountains:* A highland (alpine) climate where mean annual precipitation is 522 mm.

*Peruvian Andes:* Precipitation patterns controlled by the interaction of synoptic-scale atmospheric currents and the complex Andean topography.

*Colombian Andes:* Precipitation represented by a complex interaction of low-level jet stream with local topography and the ITCZ.

Figure 1 provides mean daily precipitation computed for all days by rain gauge elevation for the ten regions. For the Alps, the French Cévennes, and the Blue Nile regions, the rain gauges provide similar patterns and show no clear elevation-dependent pattern. For the Peruvian Andes, Colombian Andes, U.S. Rocky Mountains, Turkey, and the Himalayas we see a clear decrease in precipitation amounts with increasing elevation. Taiwan and California study sites show increasing precipitation amounts with increasing elevation. For this reason, we divided these regions into two elevation groups according to rain gauges at lower elevations (shown by black dots in the figure) and higher elevations (grey dots) throughout this study.

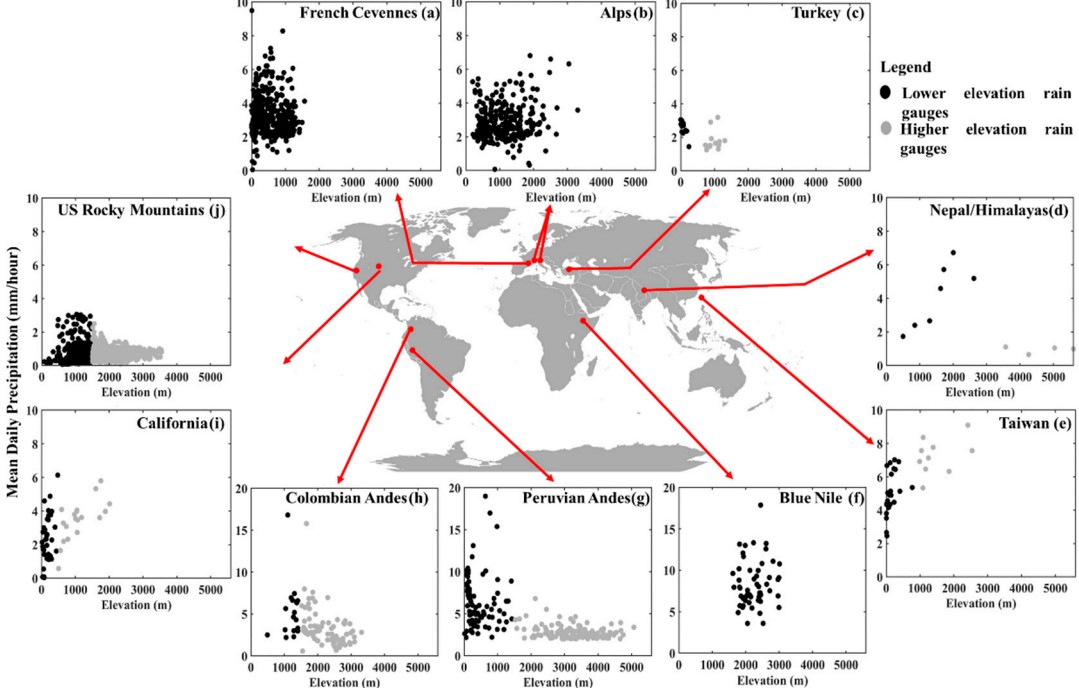

**Figure 1.** Geographic location of the study regions and mean daily precipitation (mm/hour) vs. rain gauge elevation (m) for (**a**) the French Cévennes, (**b**) the Italian Alps, (**c**) Turkey, (**d**) Nepal, (**e**) Taiwan, (**f**) the Blue Nile, (**g**) the Peruvian Andes, (**h**) the Colombian Andes, (**i**) California, and (**j**) the U.S. Rocky Mountains. Faint colors represent higher-elevation rain gauge values.

## 3. Satellite-Based Precipitation Products

The SPPs evaluated in this study are summarized in Table 2. The five SPPs are based on different techniques applied to PMW and IR measurements, which will be explained in detail in the following sections. IMERGV05B Final, IMERGV06B Final, GSMaPV07 and MSWEP SPPs are gauge corrected. In addition to comparing different products that are gauge corrected, in order to understand the performance of gauge adjustment to the satellite-only products, a comparison between satellite only products and gauge-adjusted products is performed for IMERGV06B Late and GSMaPV07 mvk which are satellite only SPPs.

**Table 2.** Summary of satellite-based precipitation products (SPPs) used in this study.

| Abbreviation | Full Name | Provider | Spatial Resolution (°) | Temporal Resolution |
|---|---|---|---|---|
| IMERGV05B | Integrated Multi-satellitE Retrievals | NASA GSFC | 0.1 | 30-min |
| IMERGV06B | Integrated Multi-satellitE Retrievals | NASA GSFC | 0.1 | 30-min |
| GSMaPV07 | Global Satellite Mapping of Precipitation | JAXA | 0.1 | hourly |
| CMORPH | CPC MORPHing technique V2 | NOAA | 0.07 | 30-min |
| MSWEPV2.2 | Multi-Source Weighted-Ensemble Precipitation | | 0.1 | 3-hourly |

### 3.1. Integrated Multi-Satellite Retrievals for GPM (IMERG)

IMERG blends SPP estimates from infrared (IR) and passive microwave (PMW) sensors [14,42]. First, precipitation estimates retrieved from PMW datasets by GPROF are calibrated using: (i) GPM Combined Radar-Radiometer (CORRA) precipitation estimates, which are derived from GPM Microwave Imager (GMI) and Dual-Frequency Precipitation Radar (DPR); and (ii) Global Precipitation Climatology Project (GPCP) monthly precipitation estimates. These PMW precipitation estimates are blended with the more frequent IR-based Precipitation Estimation from Remotely Sensed Information using Artificial Neural Networks-Cloud Classification System (PERSIANN-CCS) algorithm [43] by using CMORPH-Kalman Filter Lagrangian time interpolation [15]. The blending process relies on motion vectors derived from Climate Prediction Center IR (CPC–IR) cloud top temperature maps to produce half hourly precipitation estimates. This blending technique provides near–real time estimates at high spatial (0.1°) and temporal resolution (30 min) with quasi-global coverage. As an end product, IMERG produces the separate Early- (4 h after observation), Late- (12 h after observation), and Final- (2 months after observation) Run products.

IMERG algorithm released four different versions from 2014 to 2019 (version 3, version 4, version 5, and version 6). One of the main differences among these versions is that IMERGV03 and V04 rely on the use of successively different GPROF algorithms (V03 and V04) meanwhile IMERG V05 and V06 use GPROFV05. The one of the main differences between these GPROF versions is that, GPROFV04 and GPROFV05 included threshold precipitation rate to adjust fractional coverage. Other improvements from IMERG V03, V04 to V05 are: (i) the use Global Precipitation Climatology Project (GPCP) v2.3; (ii) inclusion of the advanced Technology microwave sounder (ATMS) dataset; (iii) dynamic calibration of PERSIANN-CCS with PMW retrievals; and (iv) removal of GPCC grid box volume adjustment to eliminate blocky gauge adjustment.

Up to IMERGV05B, the Lagrangian time interpolation scheme was computed from the IR data. In IMERGV06, these vectors are computed from Modern-Era Retrospective Reanalysis 2 (MERRA-2) and Goddard Earth Observing System model (GEOS) Forward Processing (FP) data using total column water vapor (TQV). IMERGV06B had an upgrade of full intercalibration to GPM combined instrument dataset (2BCMB) meanwhile IMERGV05B was taking shortcuts. SAPHIR estimates are incorporated into IMERG for the first time in V06 and are computed with PRPS. For both V05 and V06, input precipitation rates are thresholded at 0.03 mm/h to adjust fractional coverage. Moreover, IMERGV06B increase the maximum precipitation rate from 50 to 200 mm/h and it is no longer discrete. In this study we use IMERG version 6B Late and Final and IMERGV05B Final products. As mentioned, IMERGV06B uses TQV as the variable for deriving the motion vectors. It should be noted that being a vertically integrated quantity, TQV is reduced at higher elevations and hence has the possible negative effect of orography [13]. To investigate this issue, in this study we compared IMERGV05B Final product to IMERGV06B Final product. Moreover, in order to understand the rain gauge correction effectiveness, we compared IMERGV06B Final to IMERGV06B Late products.

### 3.2. CPC MORPHing Technique (CMORPH)

The NOAA Climate Prediction Center (CPC) MORPHing technique [15] provides quasi-global estimates of precipitation at high spatial and temporal resolution. CMORPH derives precipitation estimates from PMW-only satellite estimates, which are propagated by motion vectors derived from CPC-IR data [15]. The new version of CMORPH-v1.0 uses a fixed algorithm and homogeneous input to fix substantial inconsistencies that were present between 2003 and 2006. CMORPH provides a satellite-only version (CMORPH–RAW), a bias-corrected version (CMORPH–CRT), and a satellite-gauge blended version (CMORPH–BLD). CMORPH-RAW is adjusted using CPC unified gauge-based analysis over land and the pentad GPCP over the ocean to create CMORPH-CRT. CMORPH–CRT is then combined with the gauge analysis using the optimal interpolation technique to generate CMORPH–BLD. In this study, we used CMORPH-CRT, referring to it as CMORPH, since this product provides 8km 30min resolution.

### 3.3. Global Satellite Mapping of Precipitation (GSMaP)

GSMaP estimates are derived from PMW estimates and propagated by using IR estimates [17]. The algorithm retrieves precipitation estimates from PMW datasets; then a Kalman filter model is applied to refine the precipitation rate propagated using atmospheric moving vectors derived from CPC–IR data [44], providing hourly precipitation estimates. This product (GSMaP-MVK), calibrated using CPC, unifies gauge-based analyses of global daily precipitation. In this study, we used the latest version of GSMaP-MVK, version 07. GSMaPV07 includes the GPM/DPR database and implements snowfall estimation over high latitudes. In addition, GSMaPV07 includes an improvement of the orographic rain correction method by incorporating variable threshold for orographically-forced upward vertical motion determined based on the mean horizontal wind, using GPM/DPR observations as its database, and improving the gauge-correction method both at near-real-time and standard product. Improvements to the orographic rain correction method applied in GSMaPV07 algorithm was able to give a solution to overestimates of orographic rainfall cases over inland regions. However, it has been noted in an earlier study [11] that the separation from tall orographic rainfall and other types (other than shallow orographic rainfall) of orographic enhancement have not been detected.

### 3.4. Multi-Source Weighted-Ensemble Precipitation (MSWEP)

MSWEP merges SPP, reanalysis-based, and gauged-based precipitation products [45,46]. The product, developed specifically to overcome shortages related to the performance of SPP over mountainous, tropical, and snowmelt-driven regions, considers the gauge under-catch and orographic effects by using a Budyko-based framework and global-coverage runoff observations. The temporal variability of MSWEP is determined by weighted averaging of precipitation anomalies from seven datasets: two based solely on interpolation of gauge observations (CPC Unified and GPCC), three on SPP (CMORPH, GSMaP-MVK, and TMPA 3B42RT), and two on atmospheric model reanalysis (ERA-Interim and JRA-55). For each grid cell, the weights assigned to the gauge-based estimates are calculated from the gauge network density, while the weights assigned to the SPP and reanalysis are computed from their comparative performance at the surrounding gauges. In this study, we used MSWEPV2.2.

## 4. Evaluation Methodology

The gauge adjusted SPPs in this study are based on gauge observations from the Global Precipitation Climatology Project (GPCP) monthly rain gauge analysis, developed by the Global Precipitation Climatology Centre (GPCC) [47]. To represent the dependency of our evaluation to these gauge observations, we acquired the locations of the rain gauges used by GPCC and compared them to the locations of the rain gauges used in this study for ground validation. Table 3 provides the percentage overlap of our rain gauges with GPCC's. It should be noted that 97%, 83% and 62% of

the rain gauges for the Swiss Alps, French Cévennes and Himalayas, respectively, are used by GPCC, which means that evaluations of SPPs over these regions are not fully independent to the gauges used in the gauge adjustment of the SPPs. In the remaining sites, overlap is insignificant, and our SPPs' evaluation results can be considered independent.

**Table 3.** Percentage of rain gauges used by the Global Precipitation Climatology Centre (GPCC) over each region.

| Regions | Overlap (%) | Regions | Overlap (%) |
|---------|-------------|---------|-------------|
| Blue Nile | 7.8 | California | 13.4 |
| Colombia | 2.8 | South France | 83.3 |
| Himalayas | 62.5 | Taiwan | 0 |
| NE Italy | 37.7 | Turkey | 24.3 |
| Swiss | 97.2 | Rockies USA | 5.2 |
| Peru | 5.2 | | |

The evaluation was conducted at daily and annual time scales at the SPPs' native spatial resolution. Gauges are interpolated at the SPPs' grid cells by averaging gauge measurements within common satellite grid cells. The analysis excludes satellite grid cells with no overlapping gauges. In Table 1, percentage of pixels with multiple rain gauges is provided to give an insight into the ratio of the pixels with single rain gauges vs. multiple rain gauges.

We investigated the performance of the different products using quantitative, categorical, and graphical measures. To obtain an understanding of the performance of the SPPs over the different regions and elevation groups (representing the lower-elevation and higher-elevation rain gauges), we first focused on the mean annual precipitation values. Next, we evaluated the rain occurrences of each SPP by comparing probability density function (PDF) occurrence of daily precipitation of SPP to the rain gauges for each region. We also investigated daily detection capability by calculating normalized missed rainfall volume (NMRV) and normalized false alarm satellite rainfall volume (NFASRV) metrics for each region and each SPP. Finally, we evaluated rain rates by calculating mean relative error (MRE) and centralized root-mean-square error (CRMSE). These error metrics are represented by conditioning to the rain gauge and SPP average quantile intervals—less than 20th quantile; 20th to 40th quantile; 40th to 60th quantile; 60th to 80th quantile; 80th to 95th quantile; and greater than 95th quantile—to analyze SPP performance for different precipitation exceedance values. The equations for the performance metrics that were considered are provided below:

$$MRE = \frac{\sum(S-G)}{\sum G}, \tag{1}$$

and

$$CRMSE = \frac{\sqrt{\frac{1}{M}\sum\left[S-G-\frac{1}{M}\sum(S-G)\right]^2}}{\frac{1}{M}\sum G}, \tag{2}$$

$$NMRV = \frac{\sum[(G)|G > 0 \, \& \, S = 0]}{\sum G}, \tag{3}$$

$$NFASRV = \frac{\sum[(S)|G = 0 \, \& \, S > 0]}{\sum G} \tag{4}$$

where S and G represent SPP and rain gauge estimates, respectively, and M represents total number of days. The MRE measures the systematic error component with values greater or smaller than zero, indicating over- or underestimation, respectively. CRMSE measures the random component of error, as bias has been removed. The NMRV metric quantifies the missed rainfall volume by SPP, normalized by the total reference rainfall volume throughout the study period. The NFASRV metric measures

falsely detected rainfall volume by SPP, normalized by the total reference rainfall volume during the same period of time. A good performance of the SPP would imply low systematic error (that is, MRE), accompanied by low magnitude of the random error component (CRMSE) and accurate rainfall area detection (low NMRV and NFASRV, expressed in percent hereafter).

## 5. Results

### 5.1. Regional Evaluation

Figure 2 provides mean annual precipitation over the ten study regions for IMERGV05B Final, IMERGV06B Final, IMERGV06 Late, GSMaPV07, GSMaPV07 mvk, CMORPH and MSWEP. The straight line in the figure represents the mean annual precipitation of the rain gauges over that region, and each bar represents a SPP. The lower-elevation and higher-elevation rain gauge elevation groups are represented by darker- and lighter-colored bars, respectively, for those regions that have been subdivided.

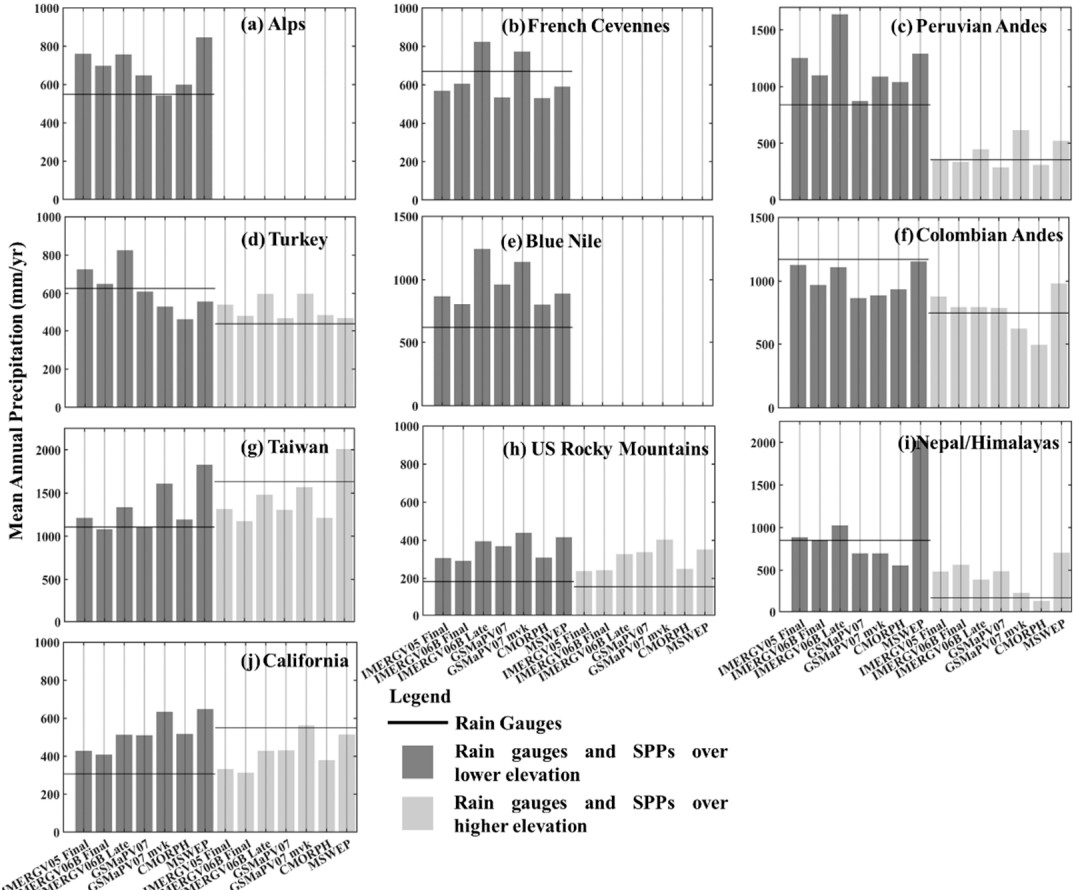

**Figure 2.** Mean annual precipitation over (**a**) the Italian Alps, (**b**) the French Cévennes, (**c**) the Peruvian Andes, (**d**) Turkey, (**e**) the Blue Nile, (**f**) the Colombian Andes, (**g**) Taiwan, (**h**) the U.S. Rocky Mountains, (**i**) Nepal, and (**j**) California for IMERGV05B Final, IMERGV06B Final, IMERGV06 Late, Global Satellite Mapping of Precipitation GSMaPV07, GSMaPV07 mvk, CMORPH and MSWEP and for rain gauges (solid black line). For regions that has elevation groups, higher elevations are represented by light grey bars and lower elevations by dark grey bars.

Over the Alps (Figure 2a; the Swiss Alps and North Italian Alps regions are combined) the mean annual precipitation measured by rain gauges is 550 mm/yr, with all the SPPs overestimating the mean annual precipitation compared to the rain gauges with GSMaPV07 mvk being an exception. CMORPH (600 mm/yr) and GSMaPV07 mvk (540 mm/yr) performed best, while MSWEP (845 mm/yr)

shows the highest overestimation over the region. It should be noted that over the Swiss Alps, 97% of the rain gauge measurements are provided to GPCC. MSWEP algorithm integrates rain gauge data from GPCC directly however this products performance was not the best compared to other SPPs. Gauge adjustment to the GSMaPV07 diminishes the performance of the product when gauge adjusted compared to satellite-only product.

Over the French Cévennes region (Figure 2b), the mean annual precipitation is 670 mm/yr, with all the SPPs underestimating except for IMERGV06B Late and GSMaPV07 mvk (satellite only SPPs). CMORPH (530 mm/yr) and GSMaPV07 (535 mm/yr) shows the highest underestimation over the region. Gauge adjustment to IMERGV06B Late and GSMaPV07 mvk turns the overestimation to slight underestimation over this region.

All the SPPs over the Blue Nile region (620 mm/yr) (Figure 2e) overestimated the annual rain gauge precipitation values significantly. The highest overestimation was by IMERGV06B Final (1200 mm/yr), followed by GSMaP mvk (1100 mm/yr). Gauge adjustment to these products significantly reduced the overestimation especially for IMERGV06B Final product. IMERGV06B Final and CMORPH exhibits the best results compared to the mean annual precipitation of the rain gauges.

Over the Peruvian Andes (Figure 2c), where the mean annual precipitation is 840 mm/yr for the lower-elevation and 350 mm/yr for the higher-elevation rain gauges, all the SPPs overestimated the lower-elevation rain gauges significantly except for GSMaPV07 (870 mm/yr). The highest overestimation over this region was by IMERGV06B Late (1630 mm/yr). This overestimation is significantly reduced with the rain gauge correction as can be seen from IMERGV06B Final. MSWEP (520 mm/yr) and GSMaPV07 mvk (610 mm/yr), significantly overestimates compared to the rest of the SPPs at higher-elevation rain gauges over this region.

Over the Colombian Andes (Figure 2f), where the mean annual precipitation is 1170 mm/yr for the lower-elevation and 740 mm/yr for the higher-elevation rain gauges, the SPPs generally underestimated the former values while slightly overestimated the latter. Over the lower-elevation rain gauges, the best performance is provided by MSWEP (1154 mm/yr), followed by IMERGV05B Late (1126 mm/yr). Over this region at lower elevations rain gauge adjustment had no improvements to the satellite-only SPPs for both GSMaPV07 mvk and IMERGV06B Late. Over the higher-elevation rain gauges, the best performance is by GSMaPV07 (780 mm/yr) and IMERGV06B Final and Late (790 mm/yr), while CMORPH (490 mm/yr) underestimated significantly compared to other products. Over this part of the region, MSWEP (980 mm/yr) overestimated the rain gauges.

Over Turkey (Figure 2d), where the mean annual precipitation is 625 mm/yr for the lower-elevation and 435 mm/yr for the higher-elevation rain gauges, SPP performances are very similar to those over the Colombian Andes. IMERGV06B Final (650 mm/yr), and GSMaPV07 (608 mm/yr) follow very closely the lower-elevation mean annual rain gauge values, while CMORPH (460 mm/yr) significantly underestimated them. Rain gauge correction over this region improved the both SPPs by reducing the overestimation. Most of the SPPs follow the mean annual precipitation values very closely over the higher-elevation rain gauges; the exceptions are IMERGV06B Late (595 mm/yr) and GSMaPv07 mvk (596 mm/yr), which overestimated the rain gauge values.

Over the U.S. Rocky Mountains (Figure 2h), where the mean annual precipitation is 182 mm/yr for the lower-elevation and 156 mm/yr for the higher-elevation rain gauges, all the SPPs overestimated the rain gauge values for both the lower and higher elevations. The performances of the individual SPPs for both elevation groups are very similar.

The performance of the SPPs over the Himalayas, where the mean annual precipitation is 850 mm/yr for the lower-elevation and 175 mm/yr for the higher-elevation rain gauges, is very similar to those provided over the Colombian Andes and Turkey. MSWEP shows significant overestimation over the both higher and lower elevation rain gauges not observed by any other SPP.CMORPH (138 mm/yr) follows the higher-elevation rain gauge (175 mm/yr) values very closely. Over the higher elevations gauge adjustment to the satellite only SPPs decreases the performance by increasing the overestimation significantly for both GSMaP mvk and IMERGV06B Late.

The MSWEP (1826 mm/yr) product over Taiwan (Figure 2g), where the mean annual precipitation is 1106 mm/yr for the lower-elevation and 1630 mm/yr for the higher-elevation rain gauges, has highest overestimation for both elevation groups. On the other hand, the SPPs were not able to capture the increase in precipitation magnitude with elevation. All the SPPs underestimated the mean annual rain gauge precipitation value significantly over the higher-elevations, with MSWEP (2005 mm/yr) again being an exception by overestimating the precipitation over this elevation group. The rain gauge adjustment to the satellite-only SPPs improve the performance of GSMaPV07 mvk and IMERGv06B Late over lower elevations meanwhile over higher elevations the performance is reduced significantly where underestimation increases compared to mean annual rain gauge values.

The SPP performances over California (Figure 2j), where the mean annual precipitation is 280 mm/yr for the lower-elevation and 760 mm/yr for the higher-elevation rain gauges, are very similar to those over Taiwan, where the SPPs are not able to capture the increase in mean annual precipitation with increasing elevations.

Overall, the SPP performances for regions that are not divided into two elevation groups (the Alps, the French Cévennes, and the Blue Nile) are very similar—that is, over the Alps and the Blue Nile they generally overestimate, and for the French Cévennes they generally underestimate. For these regions, CMORPH follows the mean annual rain gauge values very closely, while MSWEP, IMERGV06B Late and GSMaPV07 mvk overestimate the most of all the SPPs.

To summarize, the performance of MSWEP is similar to that of IMERGV06B Late and GSMaPV07 mvk, both of which are satellite-only SPPs, and MSWEP has gone through rain gauge correction. On the other hand, the IMERGV06B Final performance has improvement over lower elevation groups compared to IMERGV06B Late similar with GSMaPV07 and GSMaPV07 mvk products. The performances over the Colombian Andes, Turkey, and the Himalayas are very similar, with the SPPs underestimating (with exceptions) the lower-elevation mean annual rain gauge precipitation values and overestimating (with exceptions) the higher-elevation values. In general, the SPPs are able to capture well the decrease in precipitation amounts between the lower- and higher-elevation elevation groups. They are not, however, able to capture the increase in precipitation values between the lower- and higher-elevation groups of the Taiwan and California regions.

## 5.2. Evaluation of Rain Occurences

In this section, we report our evaluation of rain occurrence in an effort to understand the spatiotemporal variability of precipitation and the dependence of estimation errors on SPP precipitation rates. Figure 3 provides the frequency of occurrence of daily precipitation for different precipitation intensities of the rain gauges. Rainfall intensities are based on reference rainfall and shown at intervals of 0–0.5, 0.5–1, 1–2, 2–5, 5–10, 10–20, and greater than 20 mm/h. In the figure, rain gauge probability distribution is shown by grey bars, and each line represents a different SPP for each of the ten different regions. For this analysis, each region that was subdivided into two elevation groups is regrouped into one. Not shown here, but, no elevation dependence was found on the rain occurrence of the different products.



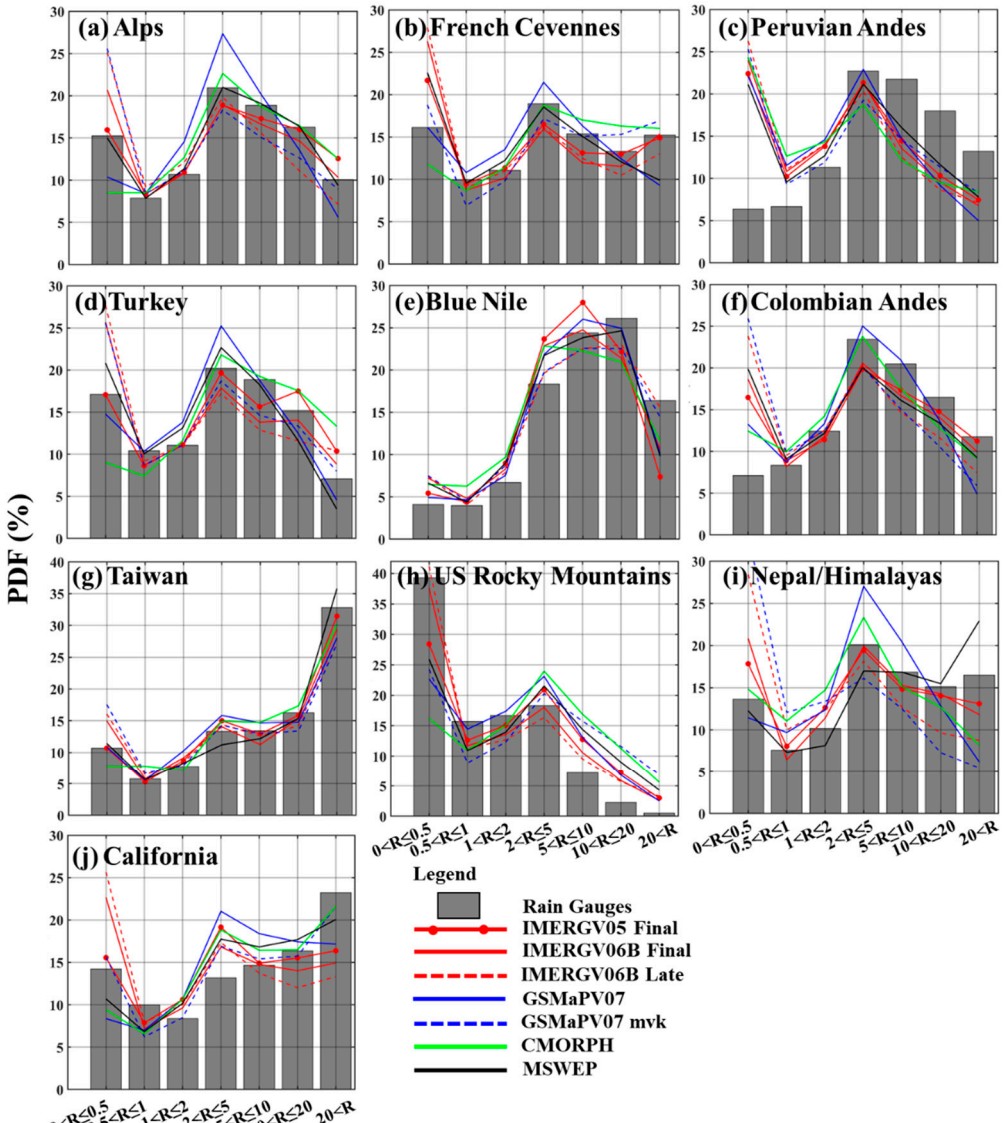

**Figure 3.** Probability density functions (PDFs) by occurrence of daily precipitation for cases with different intensities for all SPPs over (**a**) the Italian Alps, (**b**) the French Cévennes, (**c**) the Peruvian Andes, (**d**) Turkey, (**e**) the Blue Nile, (**f**) the Colombian Andes, (**g**) Taiwan, (**h**) the U.S. Rocky Mountains, (**i**) Nepal, and (**j**) California.

As Figure 3 shows, rain gauge daily precipitation distribution is, in general, very similar for the Alps, the French Cévennes, Turkey, and the Himalayas. Over these regions, occurrence frequency values of the lowest precipitation (0 < R ≤ 0.5) and moderate precipitation (2 < R ≤ 5) are most frequent than adjacent intervals. Moreover, the highest precipitation values (R > 20 and 0.5 < R ≤ 1) are less frequent compared to the rest of the intervals. The SPPs in general capture the trend of the PDF, with the exceptions.

Distributions over the Peruvian Andes, the Colombian Andes, and the Blue Nile are very similar to each other, with a clear mode observed at moderate precipitation values (2 < R ≤ 5). SPP performances for the Blue Nile are exceptional, while, on the other hand, the SPPs significantly overestimate the lowest precipitation frequencies for the Peruvian Andes and the Colombian Andes.

Over Taiwan and California, the rain gauge distributions are very similar to one another, with clear modes observed at the highest precipitation values. In general, all the SPPs over these two

regions are able to capture the trend of the rain gauge frequencies, except for IMERGV06B Final and IMERGV06B Late over the California region.

Over the Alps (Figure 3a), MSWEP captures the rain gauge distribution trend exceptionally well. As already mentioned, the region designated as "the Alps" incorporates rain gauges from the Eastern Italian Alps and the Swiss Alps. GPCC uses 97% of the rain gauges over the Swiss Alps, which could explain the exceptional performance of MSWEP over this region. The CMOPRH and GSMaPV07 products underestimate the lowest precipitation frequencies. It should be noted that GSMaPV07 has the highest overestimation in the 2–5 mm/h precipitation interval. The gauge adjustment to the GSMaPV07 mvk product turns the significant overestimation of light precipitation frequencies to significant underestimation meanwhile for the 2–20 mm/h precipitation interval gauge adjustment deteriorate the performance of the product exhibiting significant overestimation. The gauge adjustment to the IMERGV06B Late product on the other hand, reduces the significant overestimation of the light precipitation frequencies but has not much affect for the rest of the distribution. Most importantly, IMERGV05B Final product has exceptional performance capturing the rain gauge frequencies in all ranges which is significantly deteriorated when compared to IMERGV06B Final product especially for lighter precipitation frequencies.

Over Turkey (Figure 3d), IMERGV06B Final and Late overestimate the lowest precipitation range frequency and underestimate the rest of the ranges, meanwhile IMERGV05B follows the rain gauge distribution very closely in all ranges. The gauge adjustment does not provide any improvement for IMERGV06B product and its performance compared to IMERGV05B significantly low.

IMERGV06B and IMERGV05B Final products follows the rain gauge distribution trend very closely over the Himalayas (Figure 3i), with the exception of the lowest precipitation range frequency, where IMERGV06B significantly overestimates. Gauge adjustment to the IMERGV06B Late product improves the strong overestimation (underestimation) of the frequencies observed for low (high) precipitation ranges. GSMaPV07 in general follows the trend of overestimating the lowest precipitation range frequencies and underestimating the highest, while MSWEP underestimates the lower precipitation value frequencies and significantly overestimates the highest precipitation value frequencies. Gauge adjustment to the GSMaPV07 mvk product significantly improves the lower precipitation value frequencies while turns the underestimation of the medium precipitation value frequencies to an overestimation.

Over the Peruvian Andes, the SPPs are unable to capture the rain gauge precipitation frequency trend. All SPPs significantly overestimate the lowest precipitation range and underestimate the medium to high precipitation range frequencies. Over this region rain gauge correction for both products does not introduce any improvements.

Over the Colombian Andes (Figure 3f), the SPPs are similarly unable to capture the rain gauge precipitation frequency trend except CMORPH and GSMaPV07. GSMaPV07 follows the rain gauge frequency closely for the 0.5–10 mm/h ranges, showing better performance than the rest of the SPPs. The gauge adjustment to the GSMaPV07 mvk improved significantly at all precipitation value frequencies.

Over the Blue Nile (Figure 3e), all SPPs follow the low to medium range rain gauge frequency trend closely with slight overestimation of the observed values but underestimated the high precipitation range. Gauge adjustment to both products improved the performance of the SPPs over medium range precipitation value frequencies.

Over Taiwan (Figure 3g), all the SPPs follow the observed occurrence frequency distribution closely with IMERGV06B Final and Late and GSMaPV07 mvk being exception at the lighter and higher precipitation frequencies. CMORPH and IMERGV05B Final products exhibit exceptional performance over this region over all frequencies.

Overall, the SPPs are able to capture the general trend of the rain gauge precipitation frequencies over the Himalayas, Turkey, the French Cévennes, Blue Nile and the Alps. Over Peru and the Colombian Andes they are not able to capture the trend, SPPs exhibit biases at the lowest and highest precipitation ranges. The lowest range is generally problematic, especially over Peruvian and Colombian Andes,

where the SPPs significantly overestimate the frequency of the observed values from rain gauges. IMERGV05B Final product performance for lighter and higher precipitation frequencies are significantly better compared to IMERGV06B Final and Late products.

Figure 4 presents the accuracy of the products in capturing rainfall occurrence through an analysis of NMRV (blue box plots) and NFASRV (red box plots) values. The NMRV performance metric shows the rain gauge rainfall volume that the SPPs missed throughout a specific period of time, normalized by the total rain gauge rainfall volume during that period; NFASRV represents the rainfall volume that was falsely detected by the SPPs, normalized by the total rain gauge rainfall volume during the same period. In the figure, the lower-elevation rain gauges are represented by bold-colored box plots and the higher-elevation ones by faded box plots. Horizontal lines indicate the 25th and 75th percentiles and the median (red line) of the distribution; vertical lines represent the extent of the rest of the data, which is 1.5 times the 25th–75th percentile range. Outliers are represented by the "+" markers. In general, NMRV and NFASRV values are between 0% and 40% for the Alps, the French Cévennes, the Blue Nile, Taiwan, and California. NFASRV values are significantly higher for the Peruvian Andes, the Colombian Andes, and the U.S. Rocky Mountains.

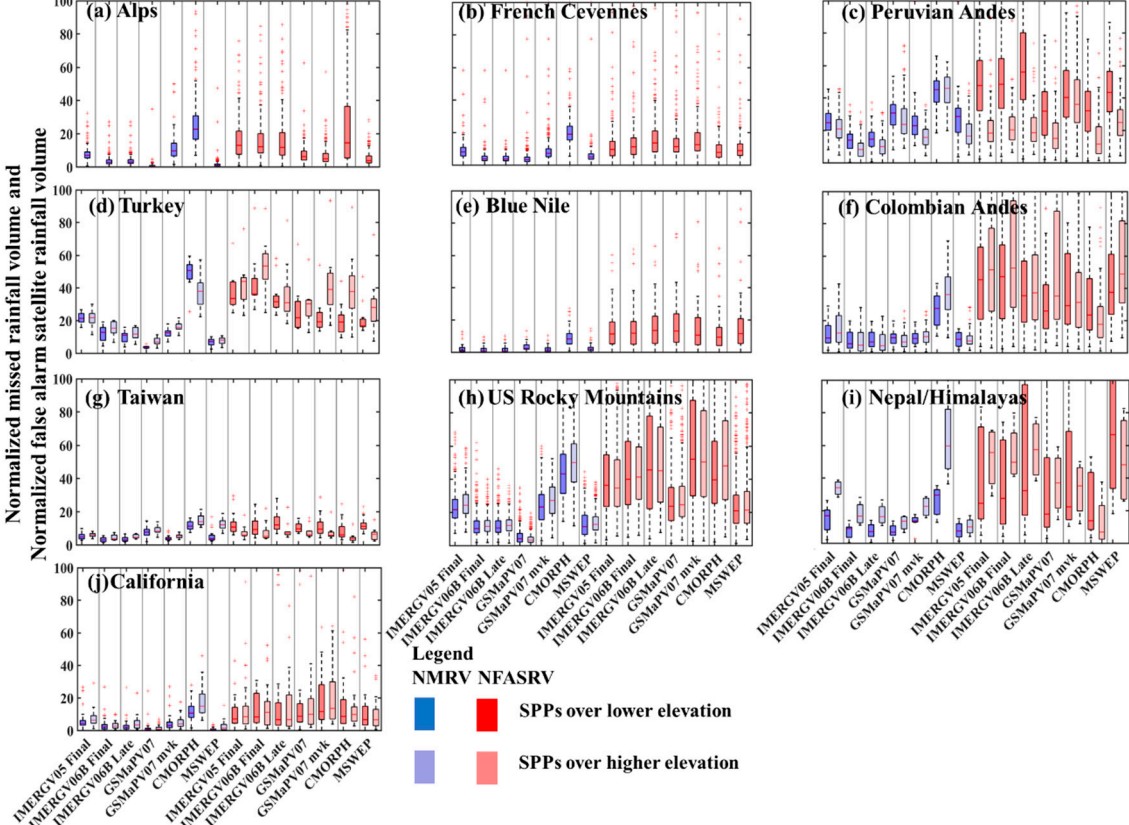

**Figure 4.** Normalized missed rainfall volume (NMRV) (blue boxplots) and normalized false alarm satellite rainfall volume (NFASRV) (red boxplots) values over (**a**) the Italian Alps, (**b**) the French Cévennes, (**c**) the Peruvian Andes, (**d**) Turkey, (**e**) the Blue Nile, (**f**) the Colombian Andes, (**g**) Taiwan, (**h**) the U.S. Rocky Mountains, (**i**) Nepal, and (**j**) California for rain gauges and every SPP (blue represents NMRV and red represents NFASRV). For regions that has elevation groups, higher-elevations are represented by faded bars.

Over the Alps (Figure 4a), the median and the variation of NFASRV are higher than NMRV values for almost all the SPPs. CMORPH has the highest median and variation of NFASRV values. Meanwhile, the gauge correction shows no improvement for both metrics. In general, IMERGV06B products shows higher NFASRV values compared to MSWEP and GSMaPV07 products.

Over Turkey (Figure 4d), the NMRV and NFASRV performances for the lower elevation rain gauges are better than the higher-elevation performances for almost all the SPPs. The worst performance of NMRV is by CMORPH for both elevations (45–55%). GSMaPV07 has the best performance in terms of both NMRV and NFASRV and both elevations. The rain gauge adjustment does not bring any improvement, especially over higher elevations. IMERGV06B Final has higher (lower) NFASRV (NMRV) values for both elevation groups compared to IMERGV05B Final product.

Over the Peruvian Andes NMRV and NFASRV median and variance values for higher-elevation rain gauges are lower compared to lowest-elevation rain gauge values. The median of the NFASRV values for lowest-elevation rain gauges reported was between 30 to 60%, and 10 to 30% for higher-elevation rain gauges. NFASRV values are significantly higher than NMRV values for all SPPs. The best performance is acheived by the IMERGV06B Late and Final products for NMRV and by GSMaPV07 for NFASRV. The rain gauge adjustment does not show any improvement for IMERGB06B Late product for both NMRV and NFASRV and for both elevations. On the other hand, GSMaPV07 mvk performance deteriorates with rain gauge adjustment for NMRV and improves for NFASRV.

Over the Colombian Andes, the medians of the NMRV values range between 5 to 35% while NFASRV median values range between 20 to 60%; hence it is clear that SPPs are falsely reporting precipitation over this region, causing an overestimation of the observed values. In general, variations of both NMRV and NFASRV are very similar for both lower- and higher-elevation rain gauges.

Over the Himalayas, the SPP performances differ slightly from each other and from their performances in other regions. In general, NMRV and NFASRV metric values are better at the lower-elevation rain gauges. CMORPH over the higher-elevation rain gauges significantly misses the precipitation values, with high medians and variances of NMRV. NFASRV variances for lower-elevation rain gauges are significantly higher than for higher-elevation gauges. MSWEP has the highest median (65%) NFASRV value over the higher-elevation rain gauges.

In general, CMORPH, GSMaPV07, and MSWEP products have the best NMRV performances over all the regions, followed by IMERGV06B Final. Depending on the region, the SPP performances on rain occurrence differ between lower- and higher-elevation rain gauges.

*5.3. Evaluation of Rain Rates*

To capture the systematic error component of SPPs, MRE is presented in Figure 5. The error statistics presented in Figures 5 and 6 were gathered by conditioning MRE and CRMSE with rain gauge and SPP average precipitation quantiles of ranges $0 < R \leq Q20$; $Q20 < R \leq Q40$; $Q40 < R \leq Q60$; $Q60 < R \leq Q80$; $Q80 < R \leq Q95$; and $Q95 < R$. Over all the regions, the SPPs move from overestimation to underestimation from the lowest precipitation quantiles to the higher quantiles, respectively, with the exception of the U.S. Rocky Mountains. The performances of all the SPPs change significantly from the lowest quantile interval to the $Q20 < R \leq Q40$ range.

Over the Alps (Figure 5a), MSWEP has the lowest MRE values for all quantile ranges. GSMaPV07 and GSMaPV07 mvk closely follows MSWEP at all quantiles, with the exception that, at the highest quantile range, these products underestimate more than MSWEP. The worst performances among the SPPs are by IMERGV05B, IMERGV06B Late and Final products, with the highest MRE values exhibited at the lowest quantiles (5) and the highest quantile (–0.8). The rain gauge correction for IMERGV06B Late increases the MRE values at lower quantiles while decreasing the MRE values at the higher quantiles. The rain gauge correction for GSMaPV07 mvk does not show any improvement.

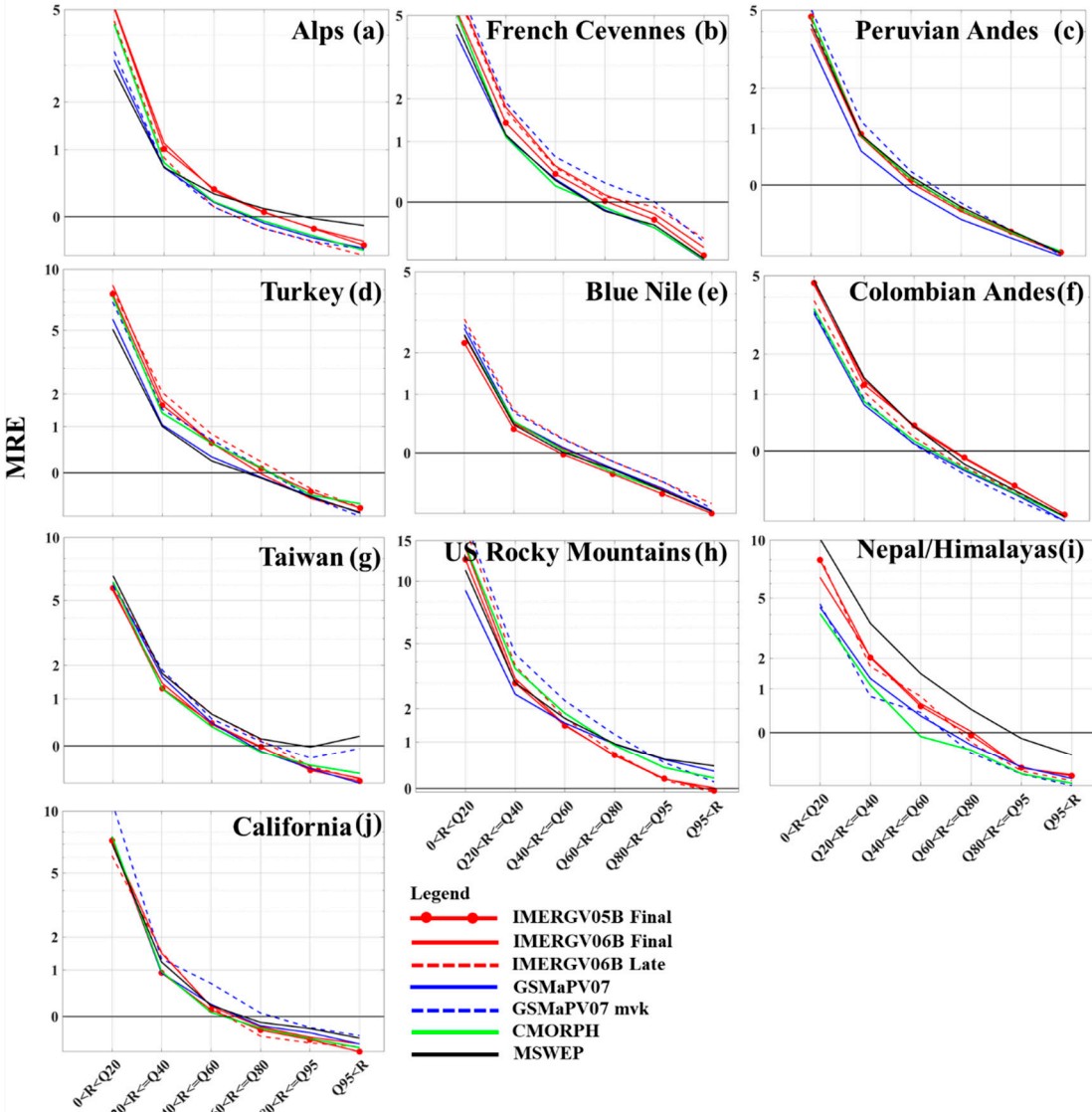

**Figure 5.** Mean relative error of daily precipitation for different quantiles of rain gauge and SPP average precipitation values for all SPPs over (**a**) the Italian Alps, (**b**) the French Cévennes, (**c**) the Peruvian Andes, (**d**) Turkey, (**e**) the Blue Nile, (**f**) the Colombian Andes, (**g**) Taiwan, (**h**) the U.S. Rocky Mountains, (**i**) Nepal, and (**j**) California.

Over the French Cévennes (Figure 5b), while GSMaPV07 displays the best performance at almost all quantile ranges, this product underestimates the most at the highest quantiles. The rain gauge correction done in GSMaPV07 mvk reduces the MRE for lower quantiles but increases the underestimation at the higher quantiles. IMERGV06B Late, Final and GSMaPV07 mvk have the worst performances, except for the highest quantile ranges, where they exhibit the lowest bias. Moreover, IMERGV05B performance for lower quantile ranges better compared to IMERGV06B.

Over the Peruvian Andes (Figure 5c), GSMaPV07 has the lowest overestimation at the lower quantile ranges and the highest underestimation at higher quantile ranges. The rain gauge correction for GSMaPV07 mvk product decreases MRE values significantly at the lower quantiles but increases MRE values at the higher quantiles slightly. IMERGV05B, IMERGV06B Final and Late product performances are identical over this region.

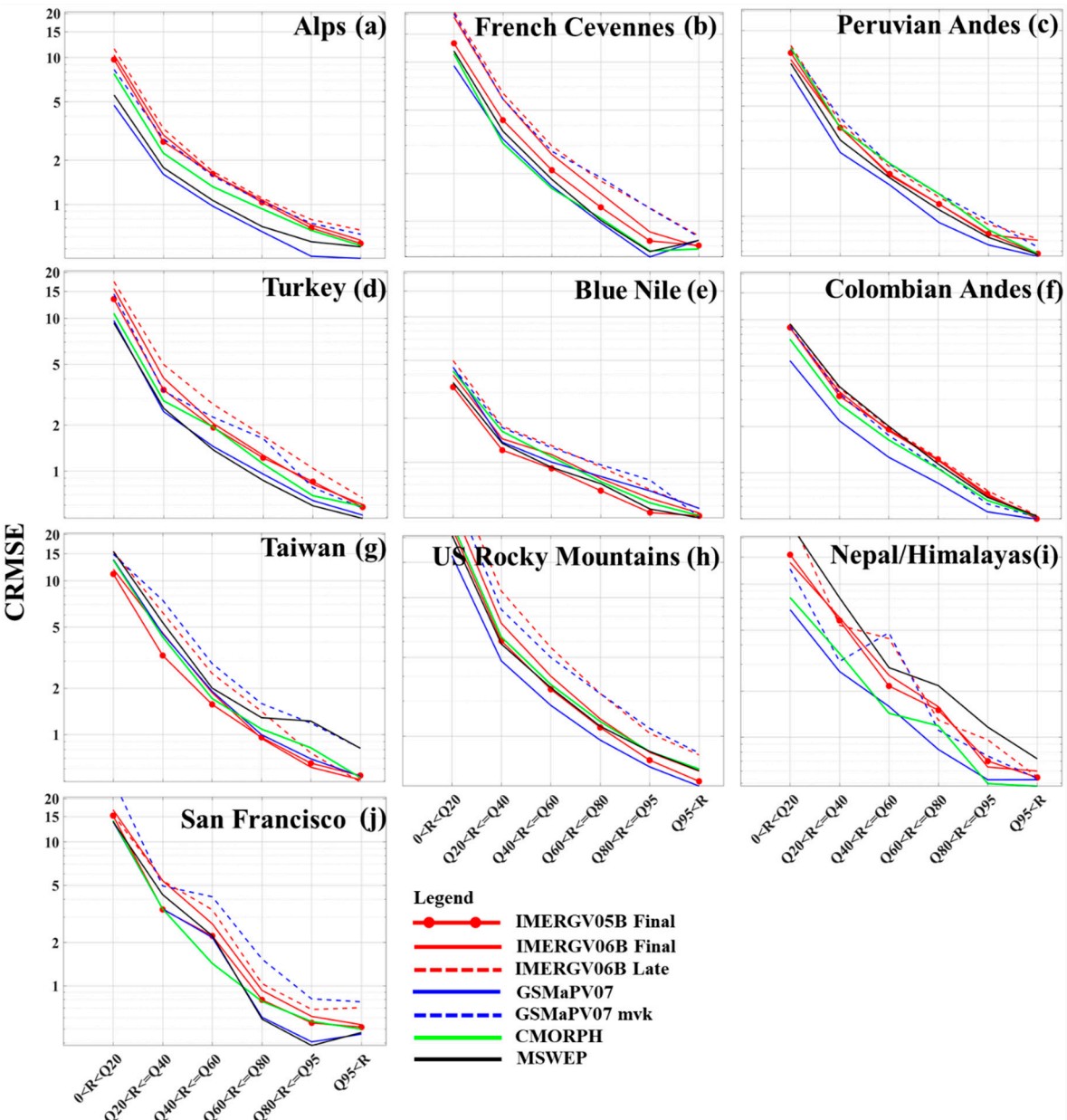

**Figure 6.** Centralized root mean square error of daily precipitation for different quantiles of rain gauge and SPP average precipitation values for all SPPs over (**a**) the Italian Alps, (**b**) the French Cévennss, (**c**) the Peruvian Andes, (**d**) Turkey, (**e**) the Blue Nile, (**f**) the Colombian Andes, (**g**) Taiwan, (**h**) the U.S. Rocky Mountains, (**i**) Nepal, and (**j**) California.

Over Turkey, the MRE values for all the SPPs range from 5 to 9 for the lowest quantile ranges. GSMaPV07 and MSWEP have the lowest overestimation (5–6) for the lowest quantile range and the highest underestimation (–0.8) for the highest. IMERGV05B, IMERGV06B Final and Late products have the highest overestimation for lower quantiles and lowest underestimation for higher quantiles.

Over the Blue Nile (Figure 5e), the MRE values range from 4 to –0.8 for all the SPPs. In general, all SPPs' systematic error patterns are very similar, with overestimation to underestimation from the lowest quantile ranges to the higher quantile ranges, respectively. The rain gauge correction decreases the overestimation at lower quantiles slightly and increases the underestimation at higher quantiles. IMERGV05B has the least overestimation for lowest quantiles compared to IMERGV06B Final and Late products.

Over the Colombian Andes (Figure 5f), CMORPH, GSMaPV07 and GSMaPV07 mvk have the lowest overestimation for all quantile ranges except the highest quantile range, where these products have the highest underestimation. MSWEP has the highest overestimation for the lower quantile ranges and high underestimation for the highest quantile ranges. Over this region the rain gauge correction conducted to IMERGV06 Late product increases the overestimation significantly for lower quantiles meanwhile has no affect for higher quantiles.

Over Taiwan (Figure 5g), MSWEP has the highest overestimation (6.5) for the lowest quantile range; on the other hand, the performance of this product is significantly better for quantiles higher than Q60 compared to the other SPPs. Rain gauge correction for IMERGV06B Late product shows no improvement when MRE values are compared meanwhile for GSMaPV07 mvk the underestimation increases significantly for higher quantiles when compared to GSMaPV07 MRE values.

Over the U.S. Rocky Mountains (Figure 5h), MRE values ranged between 15 and 0, from the lowest to the highest quantile values. GSMaPV07 had the best performance among all SPPs, except for the highest quantile range. Satellite only products (IMERGV06B Late and GSMaPV07 mvk) exhibited the highest overestimation of all SPPs and quantile ranges until Q60, after which MSWEP had the highest overestimation. The gauge correction over this region reduces the overestimation significantly.

Over the Himalayas (Figure 5i), MSWEP had the highest MRE (10) at the lowest quantile and the lowest MRE (−0.3) at the highest quantile. CMORPH, GSMaPV07 and GSMaPV07 mvk on the other hand, had the best performance at the lower quantiles. The rain gauge correction had no effect on the MRE values in this region.

Finally, Figure 6 captures the random component of error (CRMSE) of daily precipitation for the different quantiles of rain gauge and SPP average precipitation values. In general, all SPPs exhibited high CRMSE values at lower quantile ranges and lower CRMSE values at higher quantile ranges. Overall, GSMaPV07 exhibited the lowest CRMSE values over almost all regions and all quantile ranges. Moreover, IMERGV05B exhibited lower CRMSE (over some regions lowest) values compared to IMERGV06B Final and Late products over all regions. Specifically, over the Alps (Figure 6a), GSMaPV07 had the lowest CRMSE values followed by MSWEP. The rain gauge correction for GSMaPV07 mvk reduces the CRMSE values significantly for all quantile ranges. Over the French Cévennes (Figure 6b), IMERGV06B Final, Late, and GSMaPV07 mvk had the highest CRMSE values, while GSMaPV07 and CMORPH had the lowest CRMSE values. Over the Peruvian Andes (Figure 6c), GSMaPV07 had the lowest CRMSE values, while CMORPH, IMERGV06B Late and GSMaPV07 mvk had the highest. For both satellite only products the rain gauge adjustment decreased CRMSE values significantly. Over Turkey (Figure 6d), GSMaPV07 and MSWEP had the lowest CRMSE values, and IMERGV06B Late, Final, and GSMaPV07 mvk had the highest. Finally, over the Blue Nile (Figure 6e), MSWEP had the lowest CRMSE values, while IMERGV06B Late and GSMaPV07 mvk had the highest.

## 6. Discussion

This study evaluated GPM-era and TRMM-era sensors and algorithms and compared satellite only (IMERGV06B Late and GSMaPV07 mvk) SPPs to the rain gauge corrected versions in an effort to understand the improvements to them and their ongoing deficiencies, especially over complex terrain. In general, GSMaPV07 surpassed the performance of IMERGV06B Final and Late for almost all regions with respect to MRE and CRMSE values. MSWEP failed to follow the performance of the other SPPs for almost all regions with respect to occurrence frequency distributions, MRE and CRMSE values. The rain gauge correction methodology followed by the GSMaPV07 algorithm exhibited significant reduction of the CRMSE values almost over all regions, however when MRE and rain occurrences were checked we could not see improvements. The rain gauge correction methodology followed by IMERGV06B algorithm showed almost no improvements when it comes to CRMSE, MRE and rain occurrences. When it comes to different elevation groups, certain regions exhibit that NMRV and NFASRV values over lower elevation regions are slightly better compared to higher elevation regions. Moreover, there has been changes to the algorithm between IMERGV05B and IMERGV06B, as

mentioned previously. The effect of this change especially over the complex terrain is unknown and studied here. IMERGV05B showed significantly better performance capturing the lighter and heavier precipitation values compared to IMERGV06B for almost all regions. In the meantime, IMERGV05B NMRV values were worse compared to IMERGV06B and NFASRV values were similar to each other for almost all regions. More in-depth analysis should be conducted to understand the deficiency that comes with changing the method to derive motion vector with IR or TQV. However, it is obvious from this study that when motion vectors are derived using TQV, performance of detecting lighter and higher precipitation decreases over the complex terrain.

Bai and Liu [48] evaluated CMORPH, TMPA, MSWEP, and a few other SPPs over the complex terrain of the Tibetan Plateau and concluded that MSWEP generally provides the best validation results. The authors noted, however, that since MSWEP directly incorporates global-scale daily gauge data, evaluation of this product using the rain gauges could raise independence issues. In regions where GPCC overlap is minimal (i.e. Blue Nile, Colombian Andes, Peruvian Andes, and Taiwan), MSWEP performance is the worst relative to other SPPs when we consider MRE and CRMSE metrics; it has the highest overestimation for lower precipitation values and the highest underestimation for higher values. This outcome aligns with findings by Zambrano-Bigiarini and others [49], and by Nair and Indu [50]. Zambrano-Bigiarini and others [49] evaluated multiple SPPs over complex terrain over Chile, including MSWEP, CMORPH, and TMPA. They reported that MSWEP significantly overestimated lower-range precipitation amounts and underestimated higher-range amounts, while CMORPH and TMPA captured lower-range precipitation with less overestimation. Nair and Indu [50] evaluated MSWEP over India using rain gauges for pre-monsoon, monsoon, and post-monsoon seasons at daily scale for 35 years. They concluded that MSWEP shows large errors in detecting the higher quantiles of rainfall (above the 75th and 95th quantiles). Moreover, they contend that MSWEP overestimates the middle-range rainfall amounts and underestimates the higher-range amounts.

In general, the mean annual precipitation performances of the SPPs in our study are, as expected, significantly better than the daily performances. For mean annual precipitation, IMERGV06B Final and GSMaPV07 followed the values of the rain gauges closely, followed by CMORPH. On the other hand, MSWEP, IMERGV06B Final and Late showed significant overestimation.

Almost all the SPPs are able to follow the occurrence frequency of the rain gauges closely; the exceptions are MSWEP and IMERGV06B Late products. The best performance is exhibited by GSMaPV07, which can be explained by the orographic correction that the algorithm applies.

Over all the regions, SPP performance is very similar, with the SPPs overestimating the lower precipitation values significantly and underestimating the higher values. The underestimation of the higher values is in agreement with the findings of past studies [10,19,41,51] and is attributed to the occurrence of shallow, but high, accumulation precipitation related to warm-rain processes [11]. Petkovic and Kummerow [51] evaluated the performance of GPM PMW retrievals in an extreme precipitation event and concluded that satellites underestimate accumulations relative to gauges by 60%. They further concluded that fixing this problem requires better understanding of the ice content in heavy precipitation events.

Tan and others [52] evaluated IMERGv03 and TMPAV07 over the continental United States (CONUS) and concluded that IMERGV03 is better than TMPAV07 in terms of normalized mean absolute error, whereas the reverse is true for normalized RMSE. Since normalized RMSE is affected by outliers to a greater degree, this suggests IMERGV03 has more outliers, and/or the outliers have larger magnitudes. One plausible explanation for this is the fact that IMERGV03 uses a prelaunch GPM database; the transition to a full GPM database indeed improves the accuracy of IMERGV06B (with 20% decrease in MRE and nearly 4% in CRMSE) relative to TMPAV07. Similarly, Manz and others [28] evaluated IMERG Day-1 against TMPA over Peru and Ecuador and concluded that IMERG performance was worse compared to TMPA in terms of normalized RMSE for the dry coastal region of Peru.

Lastly, Maggioni and others [53] conducted an analysis related to the optimal number of rain gauges required for SPPs evaluation over complex terrain regions. They showed that the retrieval error does not exhibit strong dependence on gauge density, while the probability of no-rain detection and the mean value of missed rain volume showed dependence on gauge density.

## 7. Conclusions

We conducted an evaluation of four SPPs over ten different regions around the globe characterized by complex terrain using daily rain gauge precipitation over the 2014–2015 period. The SPPs are available at 0.1° regular latitude-longitude grids, and we averaged rain gauge measurements within those grids. The study evaluated GPM-era and TRMM-era sensors and algorithms and compared satellite only (IMERGV06B Late and GSMaPV07 mvk) to rain gauge corrected versions in an effort to understand potential improvements and ongoing deficiencies, especially over complex terrain. The study areas comprise the Alps, the French Cévennes, the western Black Sea region of Turkey, the Colombian Andes, the Peruvian Andes, the U.S. Rocky Mountains, the Blue Nile, western Taiwan, California, and the Himalayas over Nepal—all complex terrains characterized by different hydroclimatic characteristics—and we used various evaluation statistics to assess the performance of the satellite-based rainfall products. Over the Alps, the French Cévennes, the Peruvian Andes, and Turkey, we observed significant orographic effects. Taiwan, the Blue Nile, and the Colombian/Peruvian Andes are characterized by tropical conditions, and the Colombian Andes is more complex than the others, with a bimodal annual cycle of precipitation resulting from double passage of the ITCZ. The U.S. Rocky Mountains and Nepal/Himalayas can be grouped as highland climate; Nepal/Himalayas is mostly influenced by monsoon systems.

The study showed that GPM-era SPPs provided significant results in comparisons of occurrence frequencies and NMRV, NFASRV, MRE, and CRMSE values. In general, GSMaPV07 surpassed the performance of IMERGV06B Final for almost all regions with respect to MRE and CRMSE values. CMORPH surpassed the performance of MSWEP and IMERGV06 Late for almost all regions with respect to MRE and CRMSE values. Rain gauge adjustment to the IMERGV06B Late and GSMaPV07 mvk showed that the algorithms do not address the magnitude dependence of the systematic error. Meanwhile the GSMaPV07 rain gauge adjustment algorithm performs significantly well in reducing the random component of the error. As also shown by Derin et al. [19] for the TRMM-era products, the GPM-era SPP performance still depends highly on rainfall structure. Derin et al. [19] had concluded that the performances of gauge-adjusted SPPs are dependent on the representativeness of the local rain gauges. This is reflected in the limited performance improvements of gauge adjustment over satellite-only SPP products. Taking into account all the results presented in this study, we conclude the following:

(1) IMERGV06B Final and GSMaPV07 products follow the mean annual precipitation values of rain gauges very closely, followed by CMORPH, while GSMaP mvk, IMERGV06B Late and MSWEP show significant overestimation.

(2) IMERGV06B Final (–0.3–0.8 MRE; 1.1–1.9 CRMSE; 1.8–19 NMRV; 17–50 NFASRV) and GSMaPV07 (–0.1–1.1 MRE; 0.9–1.9 CRMSE; 0.9–34 NMRV; 7.6–35 NFASRV) provide error metric improvements over MSWEP (–0.3–1.3 MRE; 1.0–2.8 CRMSE; 0.9–32 NMRV; 5–93 NFASRV).

(3) The new orographic rainfall classification in the GSMaPV07 algorithm is able to improve on the detection of orographic rainfall, the rainfall amounts, and error metrics. It should be noted, moreover, that we demonstrated the improvement of this product beyond the Asian land region.

(4) IMERGV05B showed significantly better performance in capturing the lighter and heavier precipitation values compared to IMERGV06B for almost all regions. When motion vectors are derived using TQV, performance of detecting lighter and higher precipitation decreases over the complex terrain.

(5) The SPPs are able to capture the distribution of precipitation rates for all regions, although low precipitation rates are overestimated for some.

It is worth mentioning that in situ observations, while extremely important, provide only limited representation of heavy precipitation over complex terrain, and spatial-scale mismatch between satellites and gauges should be considered. Therefore, future work should consider utilizing observations from more advanced sensors—for example, X-band polarimetric radar—that provide better characterization of the spatial variability of precipitation over complex terrain and thus can enhance investigation of satellite-rainfall error characteristics [53]. Moreover, future research work in this area should include the evaluation of instantaneous PMW precipitation datasets, which constitute the basis of most combined satellite precipitation algorithms. Such analysis can give further insight into the error characteristics of SPPs. Finally, investigation of possible links between the errors and physiographic features within each region can provide more detailed error characteristics, which can provide useful information to algorithm developers and data users.

**Author Contributions:** Y.D. conceived conceptualization, methodology, data curation, software, validation, formal analysis, investigation; writing—original draft preparation, visualization. E.A. conceived conceptualization, methodology, resources, writing—review and editing; visualization; supervision; project administration; funding acquisition. G.D., M.B., B.M., E.I.N., K.K.Y., B.B. helped data curation; writing—review and editing, visualization. J.-P.R.-S., W.B., W.L.-C., F.S., H.J.V. helped data curation; writing—review and editing. A.B., H.C., D.S., S.M., C.-H.C., Y.C.H. helped data curation.

**Funding:** This work was partially supported by a NASA Precipitation Measurement Mission award (NNX07AE31G) and the Eversource Energy Center at the University of Connecticut.

**Conflicts of Interest:** The authors declare no conflict of interest. The funders had no role in the design of the study; in the collection, analyses, or interpretation of data; in the writing of the manuscript, or in the decision to publish the results.

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
