# Peer review of "Evaluation of GPM-era Global Satellite Precipitation Products over Multiple Complex Terrain Regions"

_remotesensing, doi:10.3390/rs11242936_

Round 1

Reviewer 1 Report

Review of remotesensing-628280

Products over Multiple Complex Terrain Regions

By Yagmur Derin, Emmanouil Anagnostou, Alexis Berne, Marco Borga, Brice Boudevillain, Wouter Buytaert, Che-Hao Chang, Haonan Chen, Guy Delrieu, Yung Chia Hsu, Waldo Lavado-Casimiro, Bastian Manz, Semu Moges, Efthymios I. Nikolopoulos, Dejene Sahlu, Franco Salerno, Juan-Pablo Rodríguez-Sánchez, Humberto J. Vergara and Koray K. Yilmaz

The paper evaluates the accuracy of four satellite precipitation products over high terrain regions in four continents. Two of these products were released very recently. The products were compared to daily gauge observations over a two-year period using different statistics. The topic is interesting to the research community and the research is worthwhile. However, the description of the results is awfully lengthy and need to be greatly shortened. The reviewer recommends acceptance of the paper after major revision.

Major comments:

The description on the results needs to be significantly shortened. There is an error in the citation numbers, e.g. Reference (19) is cited as (16), etc. Line 276: “hourly”????. Line 324: there is an error in the numbers. Line 328: “Final” or “Late”? Line 433: is this a probability distribution? Lines 663-665: the sentence is confusing. Needs to be rewritten “higher quantile intervals”. Line 782: “point” should replace “in situ”.

Author Response

Please see uploaded document

Reviewer 2 Report

The manuscript by Derin et al., 2019 presents an interesting study on the evaluation of satellite-based precipitation over mountainous regions. Although many evaluation studies exist in the literature, the current study put its focus on mountainous areas that are known for having significant impacts on satellite-based precipitation, particularly for microwave remote sensing. I think the study is well designed and clearly presented. I would suggest it for publication after accounting for my comments below.

The authors selected several satellite-based precipitation products. What is your selection criteria? Why not other products? For example, the CHIRPS product has also been shown to have good performance and has spatial resolution of 5 km. Please clarify it in your manuscript.

It is indeed important for validating/evaluating satellite-based precipitation at point scale using gauged measurements. However, the evaluation at large scale is also essential. I understand it is still challenging to conduct such evaluation at large scale. Nevertheless, I would suggest the authors at least show some spatial pattern comparison of all these satellite-based products. Such analysis will give us a general view of how different of these products spatially.

Author Response

Please see uploaded document.

Reviewer 3 Report

L42: GPM stands for “Global Precipitation Measurement”

Daily precipitation accumulation from rain gauges across the globe does not only vary in terms of quality control, but the daily accumulation convention also differs. How authors taken care of such discrepancy?

The basis of classification of lower elevation rain gauges and higher elevation rain gauges is not clear. In Fig. 1, the scatter plots for Himalayas and Taiwan show opposite nature. Please elaborate it.

Among four SPPs, CMORPH-CRT is not gauge corrected while others are gauge-corrected products. Hence, it does not appear a fair comparison.

I am wondering whether CMORPH and MSWEP use GPM observations? If not, how they could be GPM-era products?

Did the error metrics show similar characteristics for the grids with one rain gauges and grids with more than one rain gauges?

It would be great if authors could shed some light on the optimal number of rain gauges required for SPPs evaluation over such complex regions.

Why SPPs generally overestimate light precipitation occurrences?

Author Response

Please see uploaded document.

Round 2

Reviewer 1 Report

I believe the authors responded adequately to my comments. I suggest acceptance in present form.

Reviewer 2 Report

Thanks for your responses to my comments. I have no further comments. However, please revise a minor problem in the revision.

In the supplementary material, add the time of the image, as well as the unit in color bar. 

Please take a careful proof reading. After that,  Iw would recommend it for acceptance.

Reviewer 3 Report

The authors have addressed all comments satisfactorily and I recommend it for acceptance to the journal.